# Evaluation of Three Soil Moisture Profile Sensors Using Laboratory and Field Experiments

**DOI:** 10.3390/s23146581

**Published:** 2023-07-21

**Authors:** Felix Nieberding, Johan Alexander Huisman, Christof Huebner, Bernd Schilling, Ansgar Weuthen, Heye Reemt Bogena

**Affiliations:** 1Agrosphere Institute (IBG-3), Forschungszentrum Jülich GmbH, 52425 Jülich, Germany; 2TRUEBNER GmbH, 67435 Neustadt, Germany

**Keywords:** soil moisture profile sensors, soil water content, dielectric permittivity, sensor-to-sensor variability, temperature stability

## Abstract

Soil moisture profile sensors (SMPSs) have a high potential for climate-smart agriculture due to their easy handling and ability to perform simultaneous measurements at different depths. To date, an accurate and easy-to-use method for the evaluation of long SMPSs is not available. In this study, we developed laboratory and field experiments to evaluate three different SMPSs (SoilVUE10, Drill&Drop, and SMT500) in terms of measurement accuracy, sensor-to-sensor variability, and temperature stability. The laboratory experiment features a temperature-controlled lysimeter to evaluate intra-sensor variability and temperature stability of SMPSs. The field experiment features a water level-controlled sandbox and reference TDR measurements to evaluate the soil water measurement accuracy of the SMPS. In both experiments, a well-characterized fine sand was used as measurement medium to ensure homogeneous dielectric properties in the measurement domain of the sensors. The laboratory experiments with the lysimeter showed that the Drill&Drop sensor has the highest temperature sensitivity with a decrease of 0.014 m^3^ m^−3^ per 10 °C, but at the same time showed the lowest intra- and inter-sensor variability. The field experiment with the sandbox showed that all three SMPSs have a similar performance (average RMSE ≈ 0.023 m^3^ m^−3^) with higher uncertainties at intermediate soil moisture contents. The presented combination of laboratory and field tests were found to be well suited to evaluate the performance of SMPSs and will be used to test additional SMPSs in the future.

## 1. Introduction

The expected population growth in conjunction with an increase in frequency and intensity of (summer) droughts due to climate change increases the need for large-scale crop irrigation in many regions of the world [1]. This increasing irrigation demand will have to be met primarily through withdrawal of groundwater and freshwater resources. In addition, changing snowfall and snowmelt patterns will reduce freshwater inputs from the mountains and the degradation of wetlands further decreases water availability, possibly shifting groundwater extraction to unsustainable levels by 2050 [2]. To ensure long-term food security, drought-induced yield losses must be prevented through efficient and economical use of our limited water resources.

Real-time soil moisture monitoring is a powerful tool that allows farmers to optimize irrigation, fertilization, and tillage scheduling to conserve water, reduce nutrient leaching, and improve crop productivity and quality [3]. Using Internet of things (IoT) technology, soil moisture measurements can be integrated into wireless sensor networks (WSNs) to provide near real-time information for water- and nutrient-efficient agro-technical measures [4,5], often referred to as smart farming or agriculture 4.0 [6]. These data can be integrated into stochastic or machine learning models to forecast soil water contents under differing agricultural and climatic scenarios [7,8,9]. Due to their (perceived) simplicity, low power consumption, and non-destructive nature, electromagnetic soil moisture measurements are particularly suitable for data-driven decision making in agriculture [10,11,12]. These sensors can be assigned into different classes including time domain reflectometry (TDR), time domain transmissometry (TDT), frequency domain reflection (FDR), and capacitance sensors. These electromagnetic measurement methods are all based on the composite relative permittivity ($\varepsilon_{c}$) of the soil, which is a combination of the real (in-phase) permittivity and the imaginary (out-of-phase) permittivity [13]. The $\varepsilon_{c}$ of any porous material such as soil is a combination of the permittivities of its gaseous, liquid, and solid components [14]. As the dielectric permittivity of a material is frequency dependent, the optimal range for soil moisture determination lies between 50 MHz and 1 GHz. At lower frequencies, $\varepsilon_{c}$ depends strongly on the physical and chemical properties of the soil and above that range, it falls off due to water relaxation [13]. At 1 GHz and 25 °C, the permittivity of water and air is 78.6 and 1, respectively, whereas the permittivity of common minerals in soils and rocks is between 4 and 10 [15]. Because water has a much higher permittivity than air, the composite dielectric permittivity of the soil is closely correlated to its moisture content. However, at low measurement frequencies, the imaginary part of $\varepsilon_{c}$ is increasingly affected by the influence of bound water on soil particles with high specific surface, as well as from organic and solute content and temperature [16]. Hence, soil moisture estimates obtained with low-cost sensors operating at comparatively low frequencies often show lower measurement accuracy [17,18,19]. 

Because of their easy handling and ability to provide simultaneous measurements in different depths, so-called soil moisture profile sensors (SMPSs) exhibit high potential for climate-smart agriculture. These sensors have the advantage of being installed with minimum effort from above ground, thus keeping soil disturbance low [20]. The opportunity to apply SMPSs in agriculture 4.0 has motivated many testbed studies to evaluate how sensor performance is affected by various factors, including temperature, moisture content, soil type, soil electrical conductivity (EC), operating frequency, and sensor calibration [20,21,22,23,24,25,26]. The effects of EC and temperature changes on $\varepsilon_{c}$ are well known, and manufacturers try to compensate for these effects as much as possible. While the effect of temperature changes is usually internally corrected for, the differences in soil EC, clay content and particle shape can be addressed by soil-specific sensor calibrations. However, to date, an accurate and easy-to-use method for the evaluation of long SMPSs is not available. 

To this end, we designed a laboratory and a field experiment to better discriminate between changes in soil dielectric permittivity and sensor variability due to environmental effects. We tested three different SMPS, namely the SoilVUE10 (50 cm) from Campbell Scientific, the Drill&Drop (60 cm) from Sentek, and the SMT500 (50 cm), which is an early prototype SMPS developed by TRUEBNER GmbH. The following questions were addressed: (1) How high is the measurement variability of the vertical measurement sections of an SMPS? (2) How strong is the sensor response influenced by changes in temperature? (3) How accurately can the SMPS determine soil moisture dynamics in the root zone and how large is the sensor-to-sensor variability of the SMPS? Questions 1 and 2 were investigated by varying the temperature of a soil column with saturated sand from 5 to 40 °C. Question 3 was addressed using a 2 × 2 × 1.5 m sandbox where soil moisture was varied by controlling the water table.

## 2. Materials and Methods

### 2.1. Tested SMPSs and Measurement Principles

In the following, the SMPSs and their operating principles are presented in detail (see also Figure 1 and Table 1). 

### 2.2. The SoilVUE10 Sensor

The SoilVUE10 sensor (Campbell Scientific Inc., Logan, UT, USA) is a threaded cylindrical probe measuring $\varepsilon_{c}$, soil temperature, and soil electrical conductivity (EC) along specific depths. The probe estimates $\varepsilon_{c}$ based on the TDR concept by measuring the travel time of electromagnetic waves along a waveguide with a known length [27,28]. The TDR circuitry of the SoilVUE10 is connected to a series of six or nine helical waveguides depending on the length of the probe (55 cm or 105 cm), which provide the dielectric permittivity for different depth ranges. For every depth range, a set of three stainless steel waveguides, each 1.5 cm apart from each other, are integrated into the threaded design of the probe. In this study, we used the 55 cm version with measurements centered around depths of 5, 10, 20, 30, 40, and 50 cm [29]. The sensor has a diameter of 5.2 cm without the threads and of 5.8 cm including the threads. The sensor is screwed into the soil from the surface after pre-drilling a hole with a 5 cm auger. Thorough pre-wetting of the hole was conducted to improve the stability of the sand during installation. 

#### 2.2.1. The Drill&Drop Sensor

The Drill&Drop sensor (Sentek Pty Ltd., Stepney, Australia) is a fully encapsulated, tapered sensor measuring soil moisture, soil temperature, and electrical conductivity (optionally) in 10 cm steps along the probe. While there is a reasonable amount of literature and documentation on other profile sensors from Sentek (EnviroScan, EnviroSmart and Diviner 2000) [22,30], the documentation on the Drill&Drop is sparse. To our understanding, the probe uses FDR technology where an oscillator circuit in the sensing head determines the resonant frequency of a ring electrode, which depends on a fringe-effect capacitor and the dielectric permittivity of the soil surrounding the ring. The resonant frequency of the oscillator circuit is normalized to the scaled frequency (*SF*) expressed as:(1)$$SF=\frac{F_{a}-F_{s}}{F_{a}-F_{w}}$$
where $F_{a}$, $F_{w}$, and $F_{s}$ are the resonant frequency counts of the probe suspended in air, deionized water, and soil, respectively. The *SF* is related to soil moisture (θ) using a calibration equation:(2)$$SF=a\times \theta^{b}+c$$
where *a* (0.232), *b* (0.410), and *c* (−0.021) are soil-specific calibration equation coefficients [31,32]. In our study, we used the 60 cm version of the sensor with measurements centered in depths of 5, 15, 25, 35, 45, and 55 cm. Typically for SMPSs, the measurements are highly sensitive to air gaps between the probe and the surrounding soil material [30]. For installation, a tapered auger is used to pre-drill a hole in which the sensor fits perfectly.

**Figure 1 sensors-23-06581-f001:**
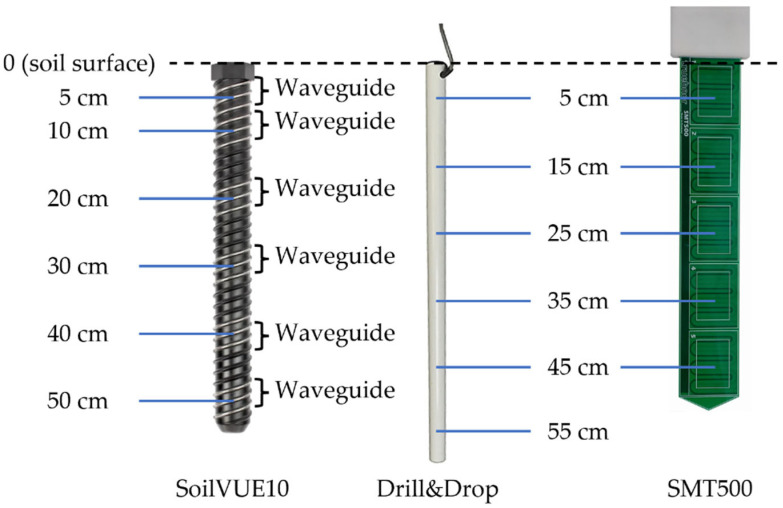
The three SMPSs evaluated in this study. From left to right: SoilVUE10 [29], Drill&Drop [33], and SMT500 (photo by Christof Huebner). Adapted with permission from [29,33], 2023, Campbell Scientific Inc., Logan, UT, USA.

#### 2.2.2. The SMT500 Sensor

The SMT500 sensor (TRUEBNER GmbH, Neustadt, Germany) is an early prototype multi-segment planar sensor using TDR technology. Since it is a new and promising sensor concept, we decided to include the SMT500 in this sensor comparison. The sensor layout and an exemplary prototype are shown in Figure 1. There are five segments in this prototype, which cover the 0–10, 10–20, 20–30, 30–40, and 40–50 cm depth ranges. The TDR electronic circuit in the head of the sensor is multiplexed to the five segments. There are major manufacturing advantages in separating the active TDR electronics in the head of the sensor from the passive electrodes in the sensor body, which allow for cost-effective production. The travel time along the TDR electrodes depends on the dielectric properties of the surrounding plastic and soil. At this point, the TDR signal analysis is simply based on voltage level thresholds to determine travel time. More advanced analysis methods are in preparation. However, it is still under development and not fully calibrated yet. An initial calibration has been performed by placing the sensor in various liquids of different dielectric properties to relate travel time to dielectric properties and water content by using the Topp equation [13]. Installation into the material can be done either by pushing in, hammering in, or with a vibratory hammer. In the case of very hard material, pre-piercing with a spring steel knife plate is also possible. Compared to tube-like SMPSs, the surface-to-cross-section ratio of the SMT500 is much higher, so that the measurement volume is large and the soil disturbance is small at the same time. 

### 2.3. Experimental Setups

For both the laboratory and the field experiment, we used a well-sieved fine sand of type F36 with an average grain size of 0.16 mm and >99% SiO_2_ content (Quarzwerke Frechen GmbH, Frechen, Germany). The selection of F36 sand was motivated by its high permeability (Ks = 2496 cm d^−1^) and absence of organic matter, which makes it easier to create a homogeneous (and hence repeatable) testbed with uniform bulk density and water content [34]. Furthermore, we can exclude the effects of salinity and lattice water on the dielectric permittivity by using an almost pure quartz sand with low electric conductivity (<0.1 dS cm^−1^).

#### 2.3.1. Laboratory Experiment

The aim of the laboratory experiment was to determine how the individual segments of the sensors react to changes in temperature and how high the variability between the segments is. To implement this experiment, one SMPS per manufacturer was installed together in a PVC tube with a diameter of 40 cm and a height of 80 cm (Figure 2). The set- up was sealed watertight at the bottom. The tube was filled with 65 cm of F36 sand on top of a 5 cm drainage layer of gravel. The gravel was separated from the sand using a drainage fleece to avoid sand clogging the pore space of the gravel. To ensure a homogeneously saturated sand body, the water was filled in from the bottom using a piezometer tube that was filtered in the drainage layer. The temperature of the tube was controlled using a heating/cooling system. For this, a 10 mm silicone hose was wrapped around the outside of the tube and the whole construction was insulated with 3.2 cm thick insulation foam (ArmaFlex, Armacell International S.A., Capellen, Luxembourg) to prevent heat loss. Water at different temperatures was pumped through the hose by means of a cryostat with a circulating bath (F3-S, Haake, Thermo Fisher Scientific Inc, Waltham, MA, USA). The water temperature was increased from 5 °C to 40 °C in 5 °C steps. Two SMT100 probes (TRUEBNER GmbH, Neustadt, Germany) were installed in the center of the sand body to determine when temperature equilibrium was reached. All sensors except the SMT500 were operated using a CR1000 datalogger (Campbell Scientific, Inc., Logan, UT, USA). The SMT500 was operated using the TrueLog system provided by the manufacturer.

#### 2.3.2. Field Experiment—The Sandbox

In the field experiment, the three types of SMPSs were installed in triplicate in a sand body (hereafter called sandbox) using the sensor arrangement shown in Figure 3. The sandbox has the dimensions of 2 × 2 × 1.5 m (L × W × D) and was installed on the experimental farm “Gut Frankenforst” of the University of Bonn, located in Königswinter, Germany. The sandbox is sealed with foil to the sides and at the bottom. A drainage layer of 20 cm gravel was used at the bottom, and the gravel layer was separated from the sand using a drainage fleece. The gravel layer was covered with 130 cm of F36 sand. Care was taken to ensure that the density of the sand body was as uniform as possible. For this purpose, the sand was filled in layers of about 10 cm thickness, whereby each filled sand layer was carefully compacted and leveled. Inhomogeneities at layer boundaries were avoided by roughening the layer surfaces after each compaction before further filling. To control the water level, four piezometers were installed in the corners of the sandbox (Figure 3), which were filtered only in the drainage layer. Three piezometers were used to accurately measure the water table changes using HYDROS21 water level sensors (METER Group, Inc., Pullman, DC, USA) installed at the bottom of each piezometer. The fourth piezometer was used to introduce water with a hose or to extract water with a pump.

TDR measurements are considered the gold standard of electromagnetic soil moisture estimation due to their high operating frequencies and corresponding low sensitivity to soil chemical and physical properties [27,35]. Therefore, a total of 18 CS610 TDR probes (Campbell Scientific, Inc.) were installed at three different positions inside the sandbox to serve as independent control measurements. These probes were sampled quasi-continuously using SDMX50 multiplexers (Campbell Scientific, Inc.) using a TDR100 system (Campbell Scientific, Inc.). The probes were horizontally installed during installation of the sandbox at depths of 5, 15, 25, 35, 45, and 55 cm. The composite dielectric permittivity $\varepsilon_{c}$ was calculated with:(3)$$\varepsilon_{c}={\left(\frac{c\Delta t_{s}}{2L}\right)}^{2}$$
where *c* is the velocity of light in free space (3 × 10^8^ m s^−1^), $\Delta t_{s}$ is the travel time of the pulse signal in the soil, and L is the length of the probe embedded in the soil [36]. Prior to the measurements in soil, the travel time in the head of the probe and the probe length were calibrated using measurements in air and demineralized water of known temperature [37]. The soil moisture content was calculated from the measured $\varepsilon_{c}$ using the empirical relationship proposed by Topp et al. [13]. 

In addition to the TDR measurements, three SoilNet units, each equipped with six calibrated SMT100 probes (TRUEBNER GmbH), were installed in the sandbox to provide additional information on soil moisture and soil temperature [38]. The SMT100 is a TDT sensor which uses a steep pulse emitted by a line driver that travels along a closed transmission line consisting of two copper strips embedded in a circuit board [39]. However, instead of measuring the pulse travel time directly as TDT sensors do, the SMT100 measures an oscillation frequency that mainly depends on the pulse travel time and thus the soil dielectric permittivity [11]. The resulting $\varepsilon_{c}$ is converted to soil moisture using the Topp et al. [13] relationship. Additionally, the SMT100 measures soil temperature using a digital temperature sensor installed in the sensor head. The probes were installed at depths of 5, 15, 25, 35, 45, and 55 cm during the filling of the sandbox. 

A CR1000X datalogger was used to operate the SoilVUE10, Drill&Drop, HYDROS 21, and TDR100 measurements and the data were stored locally. The SMT500 measurements were controlled and recorded via RS-485 using a TrueLog100 datalogger (TRUEBNER GmbH). The SMT100 sensors were controlled by the SoilNet data loggers, which wirelessly transmitted the measurement data to a server in near real-time using NBIoT technology [10].

### 2.4. Data Processing

All calculations were performed using the tidyverse family of libraries [40] for the R programming language (v. 4.2.2) [41] used within the RStudio integrated developer environment [42]. The scripts are available together with the data through a data repository. The data acquisition interval for all sensors was 10 min. As the acquisition times differed between the loggers, all measurements were aggregated to hourly mean values to ensure comparability between the systems. As the depth ranges of the SoilVUE10 differed from the other SMPS and reference sensors, the measurements were linearly inter-/extrapolated to be comparable with all other measurements, which were performed in depths of 5, 15, 25, 35, 45, and 55 cm.

#### 2.4.1. Temperature Correction

We set up the laboratory experiment to test whether the electronics of the SMPSs show any temperature sensitivity. To eliminate the effect that temperature has on the dielectric permittivity of water, we modeled $\varepsilon_{c}$ using the Complex Refraction Index Model (CRIM) [43]:(4)$$\varepsilon_{c}={\left(\theta \varepsilon_{w}^{\alpha }+\left(1-\eta \right)\varepsilon_{s}^{\alpha }+\left(\eta -1\right)\varepsilon_{a}^{\alpha }\right)}^{\frac{1}{\alpha }}$$
where η is the porosity of the soil and (1 − η), θ and (η − 1) are the volume fractions permittivities of the of the solid, aqueous, and gaseous phase, respectively. $\varepsilon_{s}$, $\varepsilon_{w}$, and $\varepsilon_{a}$ are the dielectric permittivities of the of the solid, aqueous, and gaseous phase, respectively. A porosity of 0.40 was estimated from the hanging water column experiments. Following Birchak et al. [14], the shape factor α was assumed to be 0.5. Following Robinson [44], a permittivity of 4.4 was used for the permittivity of pure quartz. The permittivity of water as a function of temperature (T in °C) was described by [43]:(5)$$\varepsilon_{w}\left(T\right)=78.54\times \left[1-4.579\times 10^{-3}\left(T-25\right)+1.19\times 10^{-5}{\left(T-25\right)}^{2}-2.8\times 10^{-8}{\left(T-25\right)}^{3}\right]$$Starting from the soil moisture measurement of each sensor, $\varepsilon_{c}$ was calculated with Equation (4) and a fixed $\varepsilon_{w}\left(T\right)$ at T = 25 °C. Then, we used the temperature that is measured from each sensor at each depth to calculate back to soil moisture using $\varepsilon_{w}\left(T\right)$. As the SMT500 does not measure soil temperature in different depths, we used the average measured soil temperature of all other available measurements. 

#### 2.4.2. Fitting van Genuchten–Mualem Parameters

The hydraulic parameters of the F36 sand were determined in triplicate using a hanging water column method. For this, samples of F36 sand were wet-packed into 100 mL steel cylinders and placed on a sandbed. The suction applied to the samples was increased stepwise, and the weight of the cylinders was determined after each pressure increase when equilibrium was reached, which lasted from several hours to multiple days depending on the water content of the sample. The applied suction and the resulting soil moisture were used to fit the van Genuchten–Mualem (vGM) parametric model [45]:(6)$$\theta =\theta_{r}+\frac{\left(\theta_{s}-\theta_{r}\right)}{{\left[1+{\left(\alpha h\right)}^{n}\right]}^{m}}$$
using a nonlinear least-squares algorithm. Here, h is the (absolute) pressure head (cm), $\theta_{s}$ and $\theta_{r}$ are the saturated and residual water content (cm^3^ cm^−3^), α (1/cm) is a parameter that is inversely related to the air-entry pressure, n is an empirical parameter related to the pore size distribution, and *m* was fixed according to:(7)m=1−1/n

The unsaturated hydraulic conductivity K(h) was calculated as:(8)$$K\left(h\right)=K_{S}S_{e}^{0.5}{\left[1-{\left(1-S_{e}^{0.5/m}\right)}^{m}\right]}^{2}$$
with a saturated hydraulic conductivity ($K_{S}$) of 2496 cm d^−1^ [34] and an effective saturation ($S_{e}$) of:(9)$$S_{e}=\frac{\theta -\theta_{r}}{\theta_{s}-\theta_{r}}$$

## 3. Results and Discussion

### 3.1. Laboratory Experiment

During the experiment, the temperature of the PVC container was increased by 5 °C every 3 to 4 days in the temperature range between 5 and 40 °C. Figure 4 shows the mean soil temperature averaged over all sensors and depths during the course of the experiment, as well as the minimum and maximum temperature at each point in time. For most of the temperature levels, the target temperature was achieved with good accuracy. Only at the highest temperature, the target temperature was not reached, likely caused by insufficient insulation of the container and tubing. This may also be the reason why larger differences between the minimum and maximum temperatures were observed at higher temperatures. On average, the temperature range for all sensors and all depths increased from 0.74 to 2.41 °C (mean = 1.29 °C). Especially the Drill&Drop probe in 5 cm depth had consistently lower temperature measurements with stronger deviation at high temperatures. When excluding that measurement, the mean range decreased to 0.91 °C, indicating an overall low variability of the temperature measurements between all sensors and depths. 

Figure 5 shows the measured soil moisture of the individual sensors at the respective depths over the course of the experiment. For better comparability, the two individual SMT100 probes are regarded as one single sensor. All sensors show a decrease in estimated soil moisture with increasing temperature, as expected from the relationship between apparent dielectric permittivity and temperature. Assuming a completely homogeneous density of the sand pack in the container, any differences in soil moisture measurements between different depths can be considered intra-sensor variability in the relationship between sensor response and permittivity. The average absolute difference (max − min) between the measurement depths of each sensor is 0.064, 0.064, 0.029, and 0.011 m^3^ m^−3^ for the SoilVUE10, SMT500, SMT100, and Drill&Drop sensors, respectively. Thus, the Drill&Drop showed the lowest intra-sensor variability of the three tested SMPSs, also lower than the variability between the two SMT100 probes. The tubular structure of the Drill&Drop might allow for a dense packing of the sand, leading to the observed low intra-sensor variability. While the sand was packed with great care, the screwed design of the SoilVUE10 might cause density variations leading to higher intra-sensor variability [21]. From the bladed structure of the SMT500, it would be expected that packing around the sensor can be performed more homogeneously than for the SoilVUE10. However, both sensors exhibit a similar intra-sensor variability indicating that the preliminary calibration of the SMT500 might need further improvement. 

The temperature sensitivity of the sensors can be evaluated from the range between minimum and maximum soil moisture values calculated for each sensor and depth. For clarity, this value is called depth-range hereafter and can be interpreted as the absolute measured soil moisture difference during the experiment. The average depth-ranges are 0.078, 0.040, 0.024, and 0.023 m^3^ m^−3^ for the Drill&Drop, SMT100, SoilVUE10, and SMT500, respectively. Hence, the SMT500 and SoilVUE10 showed the lowest temperature sensitivities over the course of the experiment.

By eliminating the effect of temperature on the composite dielectric permittivity of water using Equations (4) and (5), the remaining change in soil moisture with temperature should be related to the temperature sensitivity of the sensor electronics. To evaluate this remaining temperature dependence, we performed a linear regression analysis between the temperature corrected soil moisture estimates and temperature (Figure 6 and Table 2). As a reference, linear regression was also performed for the uncorrected measurements. To ensure homoscedasticity of the data, the slope and correlation coefficient R-squared were calculated based on the average soil moisture binned into 1 °C intervals. While the Drill&Drop and SMT100 probes exhibited a highly linear decrease in soil moisture measurements, the temperature had a more stepwise effect on the SoilVUE10 and SMT500 sensors. With an average decrease of −0.022 m^3^ m^−3^ per 10 °C before correction, the Drill&Drop showed the strongest temperature dependence of all sensors (Table 2). The temperature correction almost halved the temperature dependence of the Drill&Drop sensor to −0.014 m^3^ m^−3^ per 10 °C. The SMT500 and SoilVUE10 exhibited the lowest temperature dependence with −0.002 and −0.005 m^3^ m^−3^ per 10 °C before correction. However, the temperature correction itself did not yield lower temperature sensitivities for the SoilVUE10 and SMT500 sensors but rather changed the sign of the slope. As the $\varepsilon_{w}$ is negatively related to temperature, a positive relationship between corrected VWC and temperature indicates that the sensor electronics are also positively related to temperature. The SMT100 exhibited a low temperature dependence of −0.010 m^3^ m^−3^ per 10 °C before correction. Here, the temperature correction was able to completely remove the effect of temperature, as can be seen from the average slope of 0.000 m^3^ m^−3^ per 10 °C shown in Table 2. This indicates that the changes in soil moisture result solely from changes in $\varepsilon_{w}$ and not from the sensitivity of the sensor electronics. Our findings corroborate the results from Bogena et al. [11], who also found similar low sensitivity of the SMT100 electronics to temperature variations. In summary, the sensor electronics of the Drill&Drop showed the highest temperature sensitivity, followed by the SMT500 and SoilVUE10, while temperature effects on the SMT100 electronics can be regarded as negligible. However, it must be noted that our analysis cannot distinguish between the temperature sensitivity of the sensor electronics and the effect of a possible sensor-internal temperature correction. 

### 3.2. Field Experiment—The Sandbox

Fitting the results of the hanging water column experiments up to 130 hPa with Equations (6) and (8) yielded the soil water retention curve (SWRC) and unsaturated hydraulic conductivity curve presented in Figure 7. One of the triplicate samples for this experiment was discarded due to measurement errors. It can be seen from the figure that the transition from saturated to residual water content is very steep for the well-sorted F36 sand. The air-entry value is at 30 hPa and already at 60 hPa, the sample was almost completely drained. In the context of the sandbox experiment, this would mean that the sand above 70 cm depth would be fully drained in case of a water table at 130 cm depth. It should be noted that this is well below the maximum SMPS measurement depth of 55 cm. The F36 sand exhibits a high hydraulic conductivity of >2400 cm d^−1^ above pressures of −50 hPa, which decreases rapidly when the pressure drops below −50 hPa (Figure 7B).

A field experiment with varying water table depth was conducted between 15 September 2022 and 30 January 2023 and included both drying and wetting periods. Starting from a fully saturated sand body at the end of October, the water table was lowered to 130 cm depth and then stepwise increased by 20 to 30 cm every few weeks (Figure 8). Over the course of the experiment, the average temperature measured at the bottom of the piezometers dropped from 18.9 to 6.8 °C. However, we did not perform the temperature correction here as the drop in temperature was only moderate and the temperature sensitivity of the sensors under saturated conditions was already evaluated during the laboratory experiment. 

At the beginning of the measurement period, one of the TDR multiplexers did not work properly, resulting in occasional implausible TDR soil moisture measurements. These spikes were accounted for by removing all values <0.35 and >0.45 m^3^ m^−3^ before 26 October 2022, 10 a.m., when the multiplexer was replaced. This filtering was possible since the sandbox was fully saturated in this period. The median absolute deviation (MAD) of the three TDR measurements in every depth ranged from 0.016 to 0.17 m^3^ m^−3^, with lower values at the top and bottom depths (5, 15, and 55 cm, respectively). The intermediate depths (25 to 45 cm) showed MADs above 0.1 m^3^ m^−3^. This may be explained by the steep transition of the sand between $\theta_{r}$ and $\theta_{s}$, where small soil heterogeneities and differences in air-entry pressure already led to large differences in soil moisture. 

Figure 9 shows the relationship between all tested sensors and the reference TDR measurements. For this analysis, the mean of the triplicate TDR measurements for the respective depth ranges were considered as the reference against which each of the triplicate tested sensors were compared. With an average root mean squared error (RMSE) of 0.015 m^3^ m^−3^, the SMT100 sensors performed slightly better than the SMPSs (RMSE ≈ 0.023 m^3^ m^−3^). Only one SMT100 sensor installed at 35 cm depth consistently underestimated soil moisture, leading to overall higher RMSE of probe number 1, especially at high soil water content. This could either be a single case problem in the sensor electronics (i.e., an outlier) or it could be caused by local density variations of the sand packing. The same holds true for the SoilVUE10 probe number 1 and the SMT500 probe number 1, which consistently underestimated the soil moisture, indicating a problem with the sensor installation. If the contact between the threaded TDR circuitry of the SoilVUE10 and the soil is not optimal, measurement errors are expected to occur. Our results are consistent with Wilson et al. [21], who found that poor electrode to soil contact of the SoilVUE10 led to higher inter-sensor variability and an underestimation of the soil moisture. As the relationship between permittivity and soil moisture is nonlinear, small changes in the density of the testbed or problems during installation may lead to relatively large errors at high soil water content. In contrast, the Drill&Drop showed a remarkable consistency between the replicate measurements (Figure 10), but the saturation water content was increasingly underestimated at high soil water content as indicated by the slope of the linear regression line in Figure 9. The same holds true for the SMT500 probes, which generally underestimated the soil water content of the sandbox. It must be emphasized that we used only factory calibrations, since time and labor-intensive calibration campaigns are not performed in practice (e.g., in irrigation management). As shown in many studies, soil-specific calibration of soil moisture profile sensors will generally result in higher accuracy in soil moisture measurements [17,23,25,26]. Furthermore, the calibration of the SMT500 is preliminary and might be improved in the future.

While the overall performance of the Drill&Drop, SoilVUE10, and SMT500 is comparable, the Drill&Drop seems to exhibit non-linear behavior that leads to an overestimation in the intermediate soil moisture range. To investigate this effect further, we binned the soil moisture measurements by 1 cm water table intervals and calculated the RMSE between the SMPS and TDR measurements (Figure 10). For water tables between −50 and −100 cm, the mean RMSE of the Drill&Drop probes increased from 0.005 to 0.027 m^3^ m^−3^. For the SoilVUE10 and SMT500 sensors, this effect was also apparent, indicating that the intermediate soil moisture is most challenging to measure, probably due to the steep transition between dry and wet conditions of the well-sieved F36 sand. This may also suggest that the experiment approach is less accurate in this range, which is particularly problematic for profile sensors that do not average linearly over the measurement segments, which could be the case for the Drill&Drop. For TDR sensors such as SoilVUE10 and SMT500, this problem should be less of an issue since they provide arithmetic averages along the segments. The SMT100 sensors showed a more linear response, indicating that we were generally able to produce a vertically homogeneous sand bed. The Drill&Drop probes showed a remarkable consistency between the replicate measurements, indicating that we were also able to produce a horizontally homogeneous testbed. 

## 4. Conclusions

In this study, three electromagnetic soil moisture profile sensors were used in a laboratory experiment and in an open testbed (the sandbox) dedicated to the evaluation of temperature stability, sensor variability, and in situ soil moisture measurements. By applying a temperature correction to our laboratory measurements, we were able to exclude the effect of temperature on the soil dielectric permittivity. The remaining soil moisture variability was attributed to the temperature sensitivity of the sensor electronics. The Drill&Drop sensor showed the highest temperature sensitivity (−0.022 m^3^ m^−3^ per 10 °C), which was almost halved to 0.014 m^3^ m^−3^ per 10 °C when the change in dielectric permittivity was accounted for. With an absolute temperature sensitivity of 0.007 and 0.010 m^3^ m^−3^ per 10 °C, respectively, the SoilVUE10 and SMT500 sensors showed lower temperature stability after correction. However, both sensors exhibited higher absolute temperature sensitivities than before the correction, but with a change in sign, indicating that the sensor electronics are positively related to temperature. The SMT100 exhibited only negligible soil moisture fluctuations after correction, indicating that the sensor electronics are not susceptible to temperature variations. During the field experiment, the SMPSs were installed alongside TDR reference measurements in a 2 × 2 × 1.5 m sandbox with controlled soil moisture conditions. The well-sieved, fine sand provided a challenging environment for the SMPS evaluation. While it is well suited to producing a homogeneous testbed, the soil hydraulic conditions changed rapidly from residual to saturation water content. Hence, the three SMPSs exhibited a curvilinear response with higher RMSEs at intermediate soil moisture contents. With average RMSEs between 0.020 to 0.026 m^3^ m^−3^ per 10 °C, the overall correlations of the three SMPSs with the TDR reference measurements were quite comparable (Figure 9). The SMT100 sensors corresponded slightly better with the TDR reference measurements and showed a rather linear course of the RMSEs. As inter-sensor variability was quite low and soil moisture measurements adequate, the SMT500 might be a promising alternative to the commercially available SMPSs. However, being still under development, the overall satisfactory performance of the SMT500 should be validated for different soils and varying field conditions. Our experimental setup proved to be useful for the evaluation and characterization of soil moisture profile sensors with respect to temperature stability, inter- and intra-sensor variability, and measurement accuracy.

## Figures and Tables

**Figure 2 sensors-23-06581-f002:**
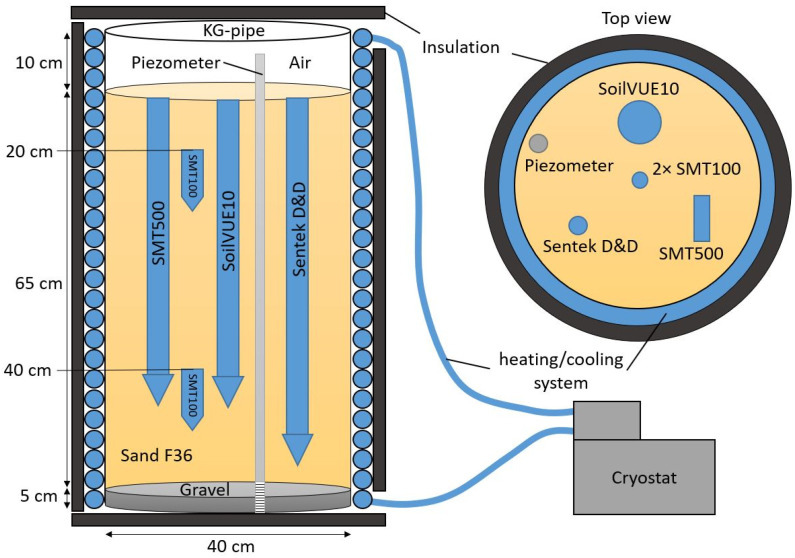
Schematic overview of the laboratory experiment.

**Figure 3 sensors-23-06581-f003:**
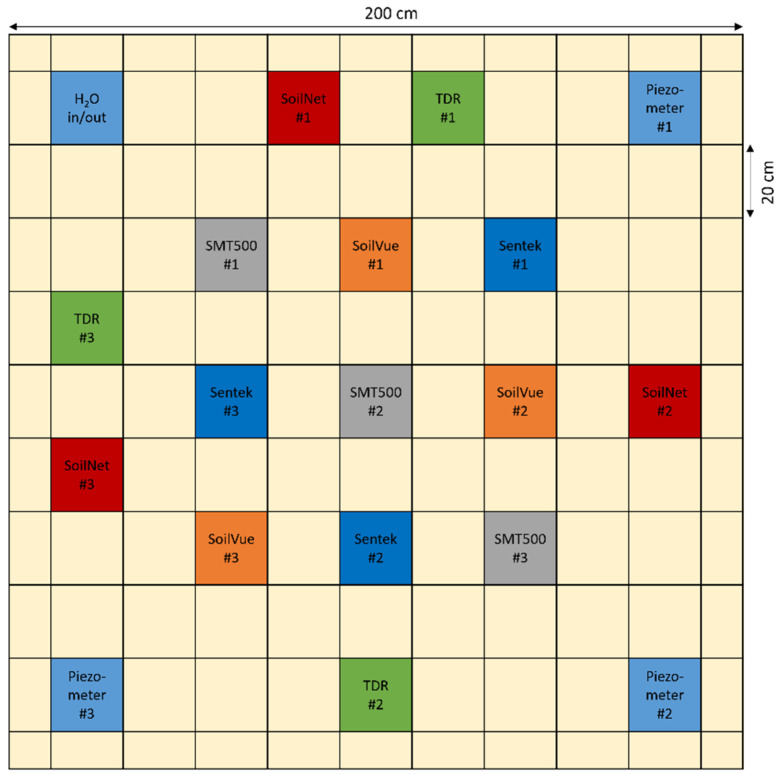
Arrangement of the sensors and piezometer in the sandbox experiment.

**Figure 4 sensors-23-06581-f004:**
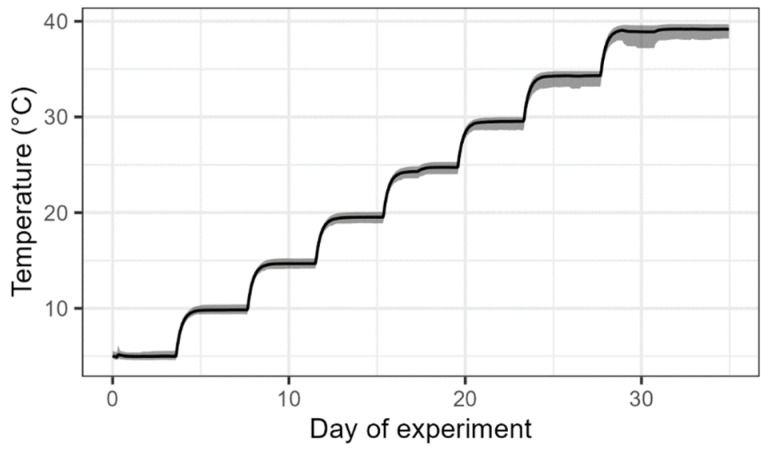
Mean measured temperature of all sensors and depths over the course of the laboratory experiment. The shaded area depicts the range between minimum and maximum temperature at each point in time.

**Figure 5 sensors-23-06581-f005:**
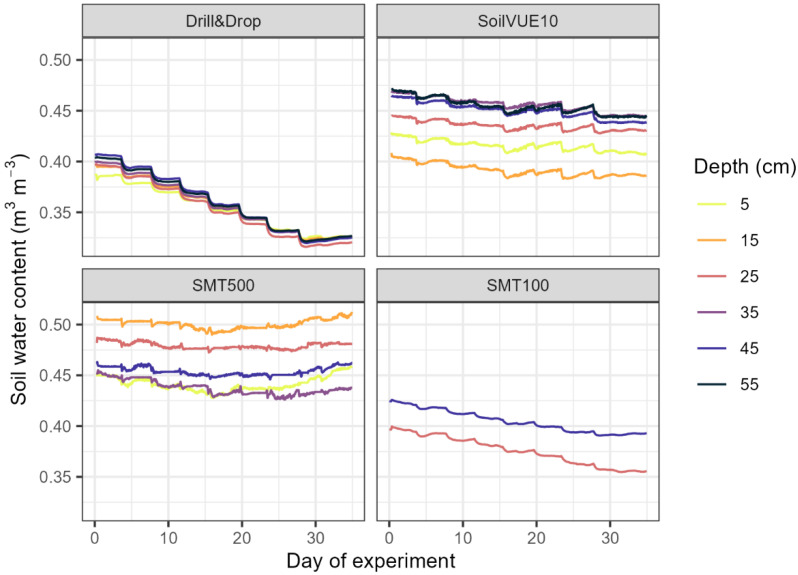
Measured soil moisture of different sensors and depths over the course of the laboratory experiment.

**Figure 6 sensors-23-06581-f006:**
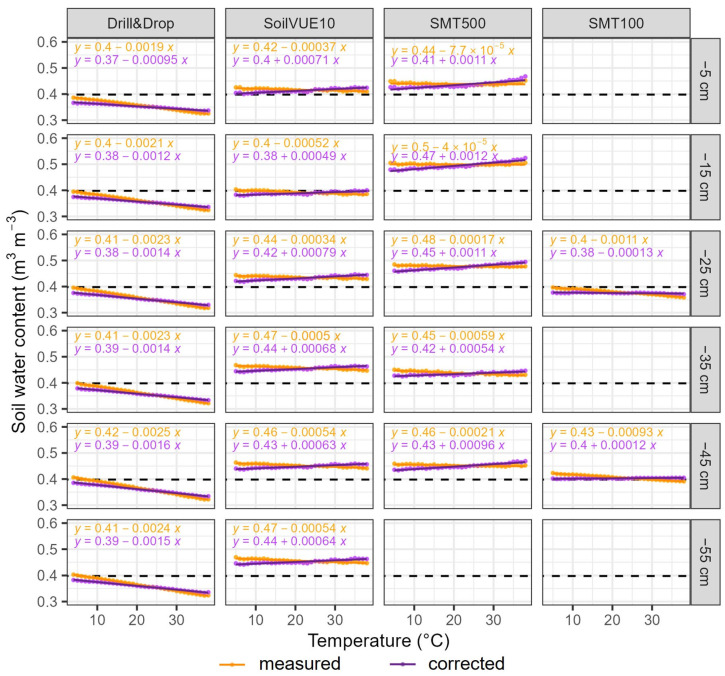
Soil moisture as a function of temperature before and after correction for the temperature effect on the dielectric permittivity of water. The soil moisture values were binned into 1 °C intervals and then averaged.

**Figure 7 sensors-23-06581-f007:**
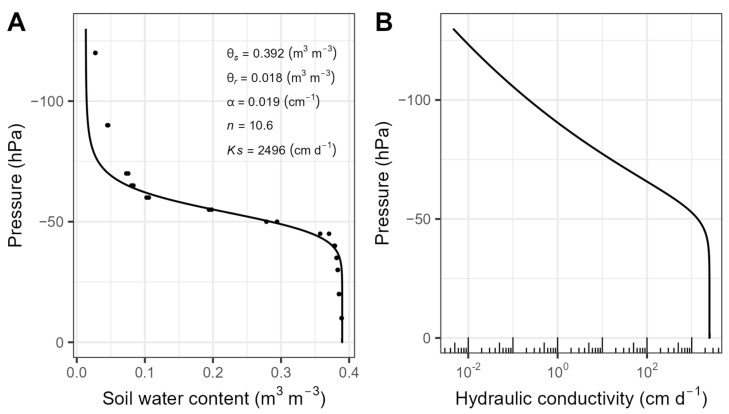
(**A**) Soil water retention curve and fitting parameters and (**B**) unsaturated hydraulic conductivity curve of F36 sand estimated with the van Genuchten–Mualem model [45]. For comparability with the sandbox, the data and curves are only shown up to a pressure of 130 hPa.

**Figure 8 sensors-23-06581-f008:**
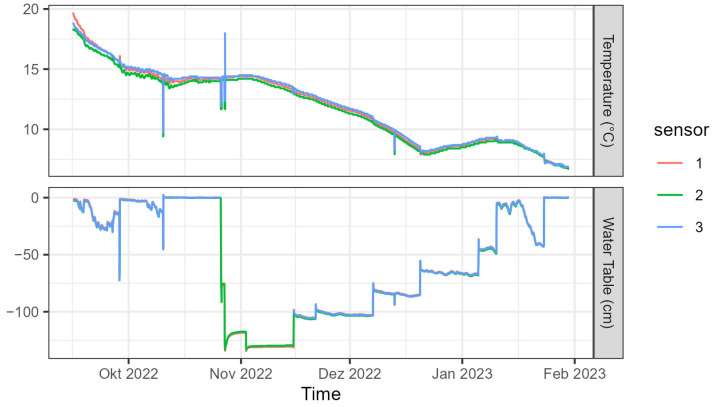
Temperature and water table measured in the piezometers of the sandbox during the field experiment.

**Figure 9 sensors-23-06581-f009:**
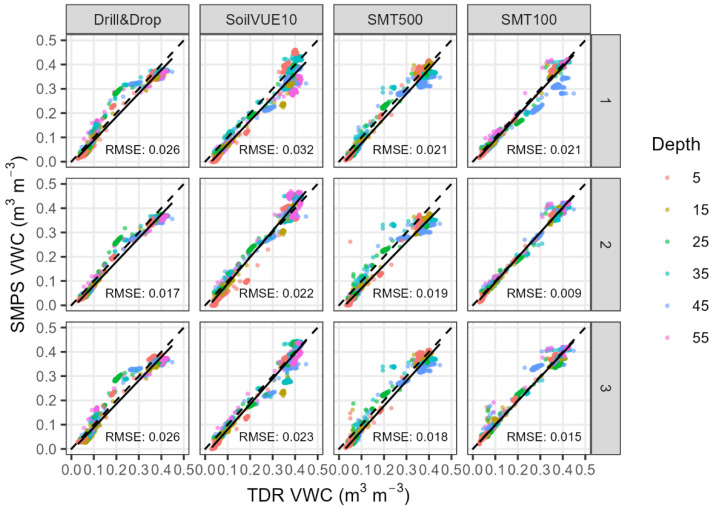
Scatter plot of soil moisture measured by all SMPSs and SMT100 sensors versus soil moisture measured by TDR for all depths; rows 1, 2, and 3 represent replicates of each sensor type.

**Figure 10 sensors-23-06581-f010:**
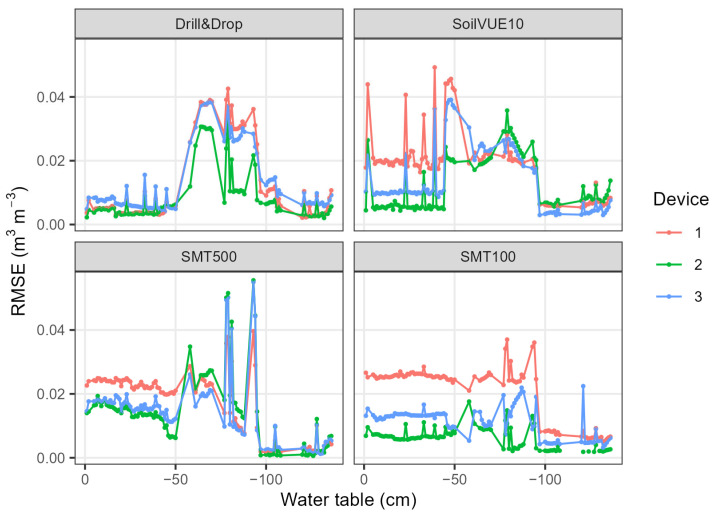
RMSE of SMPS and TDR soil water content measurements binned by 1 cm water table intervals. Devices 1, 2, and 3 represent replicates of each sensor type.

**Table 1 sensors-23-06581-t001:** Overview table of the tested SMPSs and reference measurements.

Manufacturer	Sentek	Campbell Scientific	TRUEBNER	Campbell Scientific	TRUEBNER
Sensor name	Drill&Drop	SoilVUE10	SMT500	CS610 (TDR100)	SMT100 (SoilNet)
Quantity	3	3	3	18	18
Type	Multi-depth	Multi-depth	Multi-depth	Single depth	Single depth
Measurement technology	FDR	TDR	TDR	TDR	TDT
Measured variables	T, SWC, EC (optional)	T, $\varepsilon_{c}$, SWC, EC	SWC	SWC, EC	T, $\varepsilon_{c}$, SWC
Sensor length (~cm)	60	55	50	30	20
Depth range centers (cm)	5, 15, 25, 35, 45, 55	5, 10, 20, 30, 40, 50	5, 15, 25, 35, 45	5, 15, 25, 35, 45, 55	5, 15, 25, 35, 45, 55
Net unit price (~€)	800	1600	prototype	250 + multiplexer + TDR100	100 + SoilNet

**Table 2 sensors-23-06581-t002:** Slope (m^3^ m^−3^ 10 °C^−1^) and R-squared values for linear regression between measured and temperature corrected soil moisture and temperature during the laboratory experiment for every depth and averaged over all depths (Avg.), as well as their differences.

Sensor	Depth	Slope		R-Squared		Difference	
		Measured	Corrected	Measured	Corrected	Slope	R-Squared
Drill&Drop	5	−0.019	−0.010	1.00	0.98	0.009	−0.02
	15	−0.021	−0.012	1.00	0.99	0.009	−0.01
	25	−0.023	−0.014	1.00	0.99	0.009	−0.01
	35	−0.023	−0.014	1.00	1.00	0.009	0.00
	45	−0.025	−0.016	1.00	0.99	0.009	−0.01
	55	−0.024	−0.015	1.00	0.99	0.009	−0.01
	Avg.	−0.022	−0.014	1.00	0.99	0.009	−0.01
SoilVUE10	5	−0.004	0.007	0.69	0.90	0.011	0.21
	15	−0.005	0.005	0.83	0.81	0.010	−0.02
	25	−0.003	0.008	0.71	0.93	0.011	0.22
	35	−0.005	0.007	0.87	0.93	0.012	0.06
	45	−0.005	0.006	0.89	0.92	0.011	0.03
	55	−0.005	0.006	0.81	0.86	0.011	0.05
	Avg.	−0.005	0.007	0.80	0.89	0.011	0.09
SMT500	5	−0.001	0.011	0.02	0.81	0.012	0.79
	15	0.000	0.012	0.02	0.94	0.012	0.92
	25	−0.002	0.011	0.41	0.96	0.013	0.55
	35	−0.006	0.005	0.84	0.82	0.011	−0.02
	45	−0.002	0.010	0.46	0.94	0.012	0.48
	Avg.	−0.002	0.010	0.35	0.89	0.012	0.54
SMT100	25	−0.011	−0.001	0.98	0.42	0.010	−0.56
	45	−0.009	0.001	0.99	0.70	0.010	−0.29
	Avg.	−0.010	0.000	0.98	0.56	0.010	−0.46

## Data Availability

The final data and R scripts presented in this study are openly available in the Jülich Data repository under https://doi.org/10.26165/JUELICH-DATA/NAWMCS.

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
