# Peer review of "Evaluation of Three Soil Moisture Profile Sensors Using Laboratory and Field Experiments"

_sensors, 2023, doi:10.3390/s23146581_

Round 1

Reviewer 1 Report

Comments to Authors

The author provided detailed soil moisture sensing work through their developed sensor devices. I have the following observations about the article.

Abstract: In my opinion abstract should be written in a more precise manner. Like what are the three techniques you have compared in this article? What are their significant finding, and which one have you concluded best? Thus I suggest a complete makeover of the abstract section. It just presents substantial results in the abstract section.

In the introduction section of the article, authors should add a paragraph regarding various soil moisture measurement methods like the Kirchhoff approximation model (KAM), Small Perturbation model, Integral equation model, Small slope approximation method, Dubois model, Yisok Oh model, etc. Introducing a small paragraph on these methods will create a strong base for soil moisture device development and comparison.

The sensor model, The SoilVUE10 sensor, The Drill & Drop sensor, and The SMT500 sensor must be clearly distinguished. In the first instance, it feels like all are working on the same principle. I suggest adding a comparison table to distinguish sensor models from one another. Improve the quality of Figure 1. It seems that it is copy pasted. Update this image. See the SoilVUE10 sensor, and it is getting challenging to read.

In Fig 6, why only first order fitting is performed, higher order 2nd provides better results, and the coefficient of determination of linear regression in statistics (R2) value also improves.

Table 1 modulus of the difference of Slope and R-squared can be presented so that it may be clear how much change in the parameters has occurred.

The unit of the soil water content in Fig 7 needs to be corrected. In Fig 8, specify the data for 2022 and 2023.

Additional comments

Check the equations and expressions. The unit should be rechecked. In some places, it seems quantity has become unit less.   

Minor editing of English language required

Reviewer 2 Report

 The paper deals with accurate and easy-to-use methods for the evaluation of long Soil moisture profile sensors: in order to better discriminate between changes in soil dielectric permittivity and sensor-specific variability, an experiment has been accomplished in a container filled with fine sands for evaluating three different soil moisture  profiles sensors. The paper is of interest for the community of soil scientists, dealing with soil moisture measurements. I think it can be published after some significant comments will be addressed, listed below.

-        The authors rightly refer to the use of electromagnetic soil moisture measurements, which “are particularly suitable for data-driven decision making in agriculture”: to this purpose (and this seems to be a lacking aspect in the paper) authors should mention also papers in which these measurements are used for forecasting irrigation (e.g. Giorgio et al, Sensors 2022 https://doi.org/10.3390/s22208062,  Togneri et al 2022 https://doi.org/10.1016/j.eswa.2022.117653  He et al 2022 https://doi.org/10.1016/j.agwat.2022.107618 )

- Could you provide a reference for eq (2) and comment on that? Why only a power function?

- page 3: “It has been observed that the soil sticks well to the sensor and air gaps are avoided”. How can you observe this? A comment would be appropriate.

-section 2.3 Why do not use also different types of soils, other than that kind of F36 sand, for calibrating the experiment?

- Moreover. Authors fitted the van Genuchten soil water retention curve: as far as I know, the Mualem model is generally referred to the algebraic equation relating the pressure with the unsaturated conductivity; could also author deduce this function?

- If the answer to the previous question is positive, why do authors not compare their measurement with the outcomes of Richardson-Richards’ equation for the infiltration in unsaturated soils?

- Pag 12. How can be “the water table was lowered to 130 cm depth” in the field test? Is it still a lab experiment? I do not understand: please comment.

Round 2

Reviewer 2 Report

I think authors properly replied to my comments: the paper could now be accepted for publication on Sensors.